# Human Rotaviruses of Multiple Genotypes Acquire Conserved VP4 Mutations during Serial Passage

**DOI:** 10.3390/v16060978

**Published:** 2024-06-18

**Authors:** Maximilian H. Carter, Jennifer Gribble, Julia R. Diller, Mark R. Denison, Sara A. Mirza, James D. Chappell, Natasha B. Halasa, Kristen M. Ogden

**Affiliations:** 1Department of Pediatrics, Vanderbilt University Medical Center, Nashville, TN 37232, USA; 2Department of Pathology, Microbiology, and Immunology, Vanderbilt University Medical Center, Nashville, TN 37232, USA; 3Centers for Disease Control and Prevention, Atlanta, GA 30329, USA

**Keywords:** rotavirus, VP4, genotype, serial passage, culture adaptation

## Abstract

Human rotaviruses exhibit limited tropism and replicate poorly in most cell lines. Attachment protein VP4 is a key rotavirus tropism determinant. Previous studies in which human rotaviruses were adapted to cultured cells identified mutations in VP4. However, most such studies were conducted using only a single human rotavirus genotype. In the current study, we serially passaged 50 human rotavirus clinical specimens representing five of the genotypes most frequently associated with severe human disease, each in triplicate, three to five times in primary monkey kidney cells then ten times in the MA104 monkey kidney cell line. From 13 of the 50 specimens, we obtained 25 rotavirus antigen-positive lineages representing all five genotypes, which tended to replicate more efficiently in MA104 cells at late versus early passage. We used Illumina next-generation sequencing and analysis to identify variants that arose during passage. In VP4, variants encoded 28 mutations that were conserved for all P[8] rotaviruses and 12 mutations that were conserved for all five genotypes. These findings suggest there may be a conserved mechanism of human rotavirus adaptation to MA104 cells. In the future, such a conserved adaptation mechanism could be exploited to study human rotavirus biology or efficiently manufacture vaccines.

## 1. Introduction

Rotavirus is the leading cause of diarrheal mortality for children under 5 years of age worldwide, leading to estimates of 130,000 to more than 200,000 infant and child deaths each year [1,2]. Rotaviruses cause diarrheal disease in many animal species, but exhibit narrow host and cell tropism [3,4], which has limited studies in fundamental biology and vaccine design and manufacture for human rotaviruses. A key evolutionary mechanism for RNA viruses, including rotavirus, is misincorporation of nucleotides by the viral RNA polymerase (genetic drift). Like influenza virus, rotavirus has a segmented genome, and can also reassort segments during co-infection (genetic shift) [5]. Evidence suggests there have been frequent rotavirus interspecies transmission events, sometimes with subsequent adaptation [6,7,8,9]. Together, genetic drift and shift promote evolution and can potentiate the emergence of antigenically novel rotaviruses and disease outbreaks in naïve populations [10,11]. Understanding adaptive genetic changes for human rotavirus may enhance our capacity to work with these viruses in laboratory settings or to manufacture live-attenuated vaccines at scale.

Rotavirus outer-capsid proteins determine the viral genotype. Rotaviruses are non-enveloped, triple-layered virions with a genome composed of 11 segments of double-stranded RNA [3]. The outer capsid consists of 260 VP7 glycoprotein trimers, with 60 VP4 trimers that project from the surface [12]. VP7 and VP4 determine the viral G and P type, respectively, and are the primary targets of neutralizing antibodies, which may be generated in response to infection or vaccination [13,14,15]. Although at least 42 G types and 58 P types have been defined to date, only a subset of G/P type combinations infect and cause disease in humans with varying degrees of severity ([16] and https://rega.kuleuven.be/cev/viralmetagenomics/virus-classification/rcwg accessed on 6 June 2024). Predominant human rotavirus genotypes vary by geographic region, but historically G1P[8], G2P[4], G3P[8], G4P[8], G9P[8], and G12P[8] cause the majority of human disease [17]. In recent years, several less common rotavirus genotypes, including G1P[4], G2P[8], G9P[4], G12P[4], G8P[6], G8P[8], and G12P[6], have increasing epidemiological relevance in some parts of Africa, Asia, and South America [7].

While decades of studies have yielded a wealth of knowledge about rotavirus attachment and entry [18], much remains unknown about receptor-dependent cell tropism, especially for human rotaviruses. Trimeric attachment protein VP4 primarily dictates receptor-dependent rotavirus tropism, but major outer-capsid glycoprotein VP7 can interact with integrin coreceptors to mediate internalization [19,20,21,22,23,24]. VP4 cleavage by intestinal trypsin-like proteases separates the receptor-binding ‘head’ domain (VP8*) from the stalk domain (VP5*) [23,25]. Glycans serve as attachment receptors and bind VP8* [26,27]. Many animal rotaviruses bind glycans with terminal sialic acid, but most human rotaviruses bind internal sialic acid or histo-blood group antigens, sometimes in a genotype-specific manner [18,26,28,29,30]. In the monkey kidney epithelial (MA104) cells historically used for rotavirus studies, post-attachment receptors, which interact with VP5* and VP7, contribute to cell specificity and include integrins, heat shock cognate protein hsc70, and in some cases JAM-A and occludin [18,23,24,31,32]. Differences in the capacity of monoclonal antibodies to neutralize human rotaviruses in MA104 cells and human intestinal epithelial cells suggest interactions required for entry differ among these cell types [33]. The VP5* stalk is composed of ‘body’ and ‘foot’ domains and helps mediate membrane penetration via conformational rearrangements akin to those of enveloped virus fusion proteins [23,25,34,35]. In the current model, after binding glycan receptors, VP8* head domains separate from VP5*, exposing hydrophobic loops [35]. VP5* rearranges on the surface of the infectious virus particle from an ‘upright’ to a ‘reversed’ conformation, which promotes interaction of VP5* hydrophobic loops with target membranes and outward projection of the VP5* foot that was formerly buried in the intermediate capsid layer. VP5* in the reversed information remains tethered to the rotavirus particle, inserts into membranes, and enables Ca^2+^ to cross and promote virus uncoating [35,36]. Thus, interactions of VP4 and VP7 with host molecules and factors that influence VP4 conformational rearrangements may contribute to the types of cells that rotavirus can infect and the efficiency of infection, but factors that limit or enhance human rotavirus infection in specific cell types are incompletely understood.

Human rotaviruses replicate poorly in continuous cell lines unless adapted during serial passage [37]. Consistent with the important role of VP4 in receptor-dependent cell tropism, monoreassortant rotaviruses containing human VP4, engineered using reverse genetics, replicate poorly in most cultured cells [38,39,40]. Reassortant genetics studies implicate VP4 and VP7 in pathogenesis outcomes in animals, with VP4 specifically linked to tropism [41,42,43,44,45,46]. While adaptive mutations have somewhat rarely been reported for human rotaviruses, 33 passages in African green monkey kidney (AGMK) cells yielded five non-synonymous changes in the VP4 segment of strain 89-12 [47]. These adaptive changes permitted development of the live, attenuated ROTARIX vaccine. For human rotavirus strains Wa, DC3695, and DC5685, many changes following serial passage in human colonic epithelial HT29 cells or AGMK cells arose in VP7, VP4, and NSP4 segments [48]. Passage of human rotavirus strains Wa and M in MA104 cells yielded attenuation of disease in piglets and more polymorphisms in VP4 than any other segment, including six and eight respective amino acid changes [49]. While 11 or 12 serial passages of human rotavirus vaccine candidate CDC-9 in MA104 cells resulted in no detected nucleotide or amino acid sequence changes, 28 or 44 passages in another monkey kidney epithelial cell line (Vero) resulted in five or six amino acid changes, respectively, in the VP4 gene [50]. These VP4 mutations correlated with both adaptation and attenuation, indicated by increased viral replication in cultured cells, upregulated expression of immunomodulatory cytokines, and reduced virus shedding and diarrhea in neonatal rats. Interestingly, cryo-electron microscopy revealed that at early passages, most VP4 molecules occupied the ‘reversed’ conformation on CDC-9 virus particles, which is unlikely to be capable of mediating cell entry, whereas at later passages, about half of the VP4 spikes occupied the ‘upright’ conformation associated with infectious virions [51]. It is hypothesized that an adaptive mutation in VP4 may stabilize the ‘upright’ conformation. Together, these studies further implicate VP4 in tropism and have advanced rotavirus vaccine candidates. However, all the human rotaviruses for which adaptive mutations have been reported represent a single genotype (G1P[8]), with the exception of the M strain (G3P[8]).

In the current study, we serially passaged supernatants of rotavirus-positive stool samples from pediatric patients treated at Vanderbilt University Medical Center (VUMC) between 2005 and 2013 in cultured cells. The specimens contained rotaviruses with completely sequenced genomes and represented five of the six genotypes most commonly associated with human disease, G1P[8], G2P[4], G3P[8], G9P[8], and G12P[8] [17]. High-passage rotaviruses tended to replicate more efficiently than low-passage rotaviruses, suggesting that passaged virus populations had adapted to the cells. Using next-generation sequencing and variant analysis, we identified sets of VP4 amino acid mutations that were conserved across passaged specimens and genotypes, suggesting a potentially conserved mechanism of cell culture adaptation. Conserved mutations were located primarily near the VP5* hydrophobic loops, which interact with membranes during entry, including in a residue previously identified in several other studies of rotavirus adaptation and attenuation, and near the ‘waist,’ which is adjacent to the VP7 layer in the reverse conformation. In some cases, these mutations might influence the stability of the upright conformation of VP5* on the particle. These findings help generate new hypotheses about conserved mechanisms by which rotavirus can overcome tropism barriers and replicate efficiently in cultured cells.

## 2. Materials and Methods

### 2.1. Rotavirus-Positive Clinical Specimens

Fecal specimens were collected from eligible children presenting with acute gastroenteritis at the Monroe Carell Junior Vanderbilt Children’s Hospital and Clinics during the years 2005 to 2013, and complete rotavirus genomes from rotavirus-positive specimens were previously sequenced as described [52,53,54]. In all cases, collection was performed in accordance with New Vaccine Surveillance Network (NVSN) protocols approved by the Center for Disease Control and Prevention (CDC), VUMC, and the Institutional Review Board. Informed consent, including future specimen use, was provided by a parent or guardian at the time of enrollment. For the current study, we selected 50 specimens representing five genotypes (G1P[8], G2P[4], G3P[8], G9P[8], G12P[8]). The abbreviated strain name, genotype, and year of collection for each specimen used in the current study are listed in Appendix A.

### 2.2. Cells

Primary rhesus (*Macaca mulatta*) monkey kidney (RhMK) cells (Diagnostic Hybrids, Inc., Athens, OH, USA, Cat # 49-0600A) were initially grown in 16 mm glass roller tubes in Eagle’s minimum essential medium (EMEM) supplemented with HEPES, fetal bovine serum (FBS), SV5/SV40 antisera, and gentamicin at concentrations proprietary to the manufacturer, as shipped. MA104 and Vero cells were purchased from the American Type Culture Collection (ATCC). We use the lot-specific validation criteria, including cytochrome C oxidase I gene analysis, provided by ATTC, together with visual inspection of cell morphology, culture conditions, and virus susceptibility to validate cell identity. MA104 cells and Vero cells were cultured in EMEM with Earle’s salts and L-glutamine (Corning, Corning, NY, USA) supplemented to contain 5% FBS (Gibco, Grand Island, NY, USA). Cells were cultured in serum-free media during serial passages as described in Section 2.4. All cells were maintained at 37 °C in 5% CO_2_. Primary RhMK cells in roller tubes were incubated with slow rotation. Cells were tested for mycoplasma regularly by PCR.

### 2.3. Enzyme-Linked Immunosorbent Assay (ELISA)

The Rotaclone (Meridian Bioscience, Inc., Cincinnati, OH, USA) ELISA was used to detect rotavirus in lysates from RhMK, MA104, and Vero cells according to manufacturer instructions. Samples with absorbance units (A_450_) of 0.1 or greater were considered positive.

### 2.4. Serial Passaging of Clinical Rotavirus Specimens

For an initial passage in primary RhMK cells (P1), 0.1 mL of a 10% (*w*/*v*) homogenate of stool suspension in Earle’s balanced salt solution (Sigma, Burlington, MA, USA) in triplicate for a rotavirus-positive clinical specimen was clarified by centrifugation at 10,000× *g* for 15 min. Clarified supernatants, medium alone, and a laboratory stock of SA11-4F rotavirus (0.1 mL at 5 × 10^6^ PFU/mL), were activated with 10 µg/mL trypsin (Worthington Biochemical Corporation, Lakewood, NJ, USA; LS003708) for 1 h at 37 °C then diluted in serum-free EMEM to a final trypsin concentration of <2 μg/mL. Primary RhMK cells were washed three times with serum-free EMEM and adsorbed in roller tubes with each activated specimen for 1 h at 37 °C with slow rotation. Following absorption, inocula were removed, monolayers were washed, and fresh serum-free EMEM containing 0.5 μg/mL of trypsin was added. Cells were incubated with constant rotation at 37 °C for up to 7 days or until cytopathic effect (CPE) was visible, and the cell monolayer was disrupted due to lysis. Cells then were subjected to three rounds of freezing at −80 °C and thawing prior to storage at 4 °C. In two to four subsequent passages, 0.2–1 mL of lysate from each previous lineage and passage was activated with 1 µg/mL trypsin and used as the inoculum for adsorption. The presence of rotavirus in lysates was determined by ELISA following passages one, three, and/or five. For passages subsequent to ELISA, a 1 mL inoculum was used for lysates with values <0.2; 1 mL of a 1:2.5 diluted inoculum was used for lysates with values between 0.2 and 1; and 1 mL of a 1:5 diluted inoculum was used for lysates with values >1, if additional passages were conducted.

After three to five passages in primary RhMK cells, 0.5 mL of rotavirus-positive lysates were activated with 1 μg/mL trypsin for 1 h at 37 °C. Confluent MA104 or Vero cell monolayers in T25 flasks were washed with serum-free EMEM and adsorbed with 0.5 mL (MA104) or 0.3 to 1 mL (Vero) of P3 or P5 RhMK cell lysates for 1 h at 37 °C with occasional rocking. Following absorption, inocula were removed, monolayers were washed, and fresh serum-free EMEM containing 0.5 μg/mL of trypsin was added. Cells were incubated at 37 °C for up to 7 days or until CPE was visible. Cells then were subjected to three rounds of freezing at −80 °C and thawing prior to storage at 4 °C. In up to nine subsequent passages, lysate from each previous passage was activated with 1 µg/mL trypsin and used as the inoculum for adsorption. The presence of rotavirus in lysates was determined by ELISA, typically following P3, P6, and P10. For passages subsequent to ELISA, a 1 mL inoculum was used for lysates with values <1; 1 mL of a 1:10 diluted inoculum was used for lysates with values between 1 and 2; and 1 mL of a 1:100 diluted inoculum was used for lysates with values >2. For rotavirus passages attempted in Vero cells subsequent to passage in MA104 cells, the initial inoculum was 1 mL of rotavirus-positive lysate from MA104 passage 6 or 10 that had been activated with 1 μg/mL trypsin for 1 h at 37 °C.

### 2.5. Replication Time Course

MA104 cells (~1.7 × 10^5^/well) were seeded in 24-well plates and incubated at 37 °C until confluent. Rotavirus-positive P3 or P10 lysates were diluted to 2.5 × 10^4^ FFU/mL in serum-free EMEM, to achieve a multiplicity of infection (MOI) of 0.01 fluorescent focus units (FFU)/cell with a 0.1 mL inoculum, assuming ~2.5 × 10^5^ MA104 cells per well. Lysates were used neat if titer was less than 2.5 × 10^4^ FFU/mL. Diluted rotavirus-positive lysates were activated with 1 µg/mL trypsin for 1 h at 37 °C. Cells were washed twice and adsorbed with activated viruses for 1 h at 37 °C. Cells were washed to remove unbound virus and incubated with serum-free medium plus 0.5 µg/mL trypsin at 37 °C for 0, 24, or 48 h. Plates were frozen at −80 °C and thawed three times prior to determining virus titer by fluorescent focus assay on MA104 cells. Virus yield was determined by dividing titer at 24 h or 48 h by titer at 0 h.

### 2.6. Fluorescent Focus Assay (FFA)

MA104 cells (~1 × 10^5^/well) were seeded in black-walled, clear-bottom, 96-well plates and incubated at 37 °C overnight. Virus was activated with 1 μg/mL trypsin for 1 h at 37 °C and serially diluted 1:10 in serum-free EMEM. Following two washes, cells were adsorbed with virus dilutions for 1 h at 37 °C. Cells were washed and incubated at 37 °C for 16–18 h prior to methanol fixation. Cells were stained to detect nuclei using DAPI (Invitrogen, Carlsbad, CA, USA) and rotavirus proteins using sheep α-rotavirus polyclonal serum (Invitrogen) prior to imaging and quantification using an ImageXpress Micro XL Widefield High-Content Analysis System (Applied Biosystems, Waltham, MA, USA). Virus titer was quantified from total and infected cells quantified in four fields of view/well. To determine whether P3 and P10 yield differed at 24 h or 48 h, we used two-way ANOVA followed by Šídák’s multiple comparisons test. Statistical analyses were conducted using GraphPad Prism 9.

### 2.7. RNA Extraction, RT-PCR, and Nucleotide Sequencing

Rotavirus-positive culture lysates were processed for RNA extraction, library preparation, and RNA sequencing. Two × 0.25 mL aliquots of each of MA104 P10 lysate were treated with 1 µL of DNase I (New England Biolabs, Ipswich, MA, USA) for 30 min at 37 °C then with EDTA to a final concentration of 5 mM to inactivate DNase I prior to RNA extraction using TRIzol LS (Invitrogen) according to manufacturer instructions. RNA pellets were resuspended in RNase-free water and incubated in a heat block set at 55 °C for 5–10 min, with small aliquots set aside for RNA quantitation by Qubit. RNA library preparation for Illumina sequencing was conducted using 5 µL of input RNA and the NEBNext Ultra II RNA Library Prep Kit for Illumina (New England Biolabs), according to the manufacturer’s instructions. Briefly, RNA was fragmented prior to first-strand and second-strand synthesis, AMPure XP Bead clean up, and end repair. PCR enrichment of adaptor ligated DNA was conducted using NEBNext Multiplex Oligos for Illumina (New England Biolabs). Illumina-ready libraries were sequenced by paired-end sequencing (2 × 150) on a NovaSeq 6000 Sequencing System (Illumina, San Diego, CA, USA). Assistance with quality control and next-generation sequencing was provided by the Vanderbilt Technologies for Advanced Genomics (VANTAGE) research core.

### 2.8. Illumina RNA-Seq Data Analysis and Variant Calling

Raw reads were processed by first removing the Illumina TruSeq adapter using Trimmomatic default settings [55]. Reads shorter than 36 bp were removed and low-quality bases (Q score  <  30) were trimmed from read ends. The raw FASTQ files were aligned to rotavirus reference genome segments (Appendix A) using Bowtie2, with the following parameters: bowtie2-p 32-q-x {ref}-1 {sample}_R1_paired.fastq-2 {sample}_R2_paired.fastq-U {sample}_R1_unpaired.fastq,{sample}_R2_unpaired.fastq-S {sample}_bowtie2.sam [56]. The SAMtools [57] suite was used to calculate read depth at each genomic coordinate. LoFreq [58] was used to call single nucleotide variants and indels with the following parameters: lofreq call-parallel--pp-threads 32-f {ref}-d 100000-o {sample}.vcf {sample}_bowtie2.sort.bam. Variants were filtered at a threshold frequency of 0.001, consistent with previous reports [59]. Parsing of variants common across samples was performed via the command line. Identification of encoded amino acid mutations based on published open reading frames and computationally identified variants was performed manually. Putative locations of variants were visualized using UCSF Chimera, developed by the Resource for Biocomputing, Visualization, and Informatics at the University of California, San Francisco, with support from NIH P41-GM103311 [60].

## 3. Results

### 3.1. Human Rotaviruses of Different Genotypes Can Adapt to Replication in Monkey Kidney Cells

To identify polymorphisms that enhance replication in cultured cells, we serially passaged supernatants of rotavirus-positive stool samples from pediatric patients treated at Vanderbilt University Medical Center [61]. Each patient stool sample was considered a ‘specimen’ and contained rotavirus with a sequenced genome [54]. We attempted to adapt 50 specimens to primary rhesus monkey kidney (RhMK) cells in glass roller tubes. While adaptation historically has been done in primary AGMK cells [48,62], these primary cells were unavailable commercially. The initial inocula were 0.1 mL of trypsin-activated, clarified 10% stool homogenates of rotavirus-positive clinical specimens, each in triplicate parallel ‘lineages’ (Figure 1). In subsequent passages, we used 0.2–1 mL of lysate as inoculum, based on ELISA score. We also passaged simian laboratory strain SA11 and medium containing trypsin as controls. Following up to five passages in primary RhMK cells, 35 lineages representing 18 distinct specimens tested positive for rotavirus antigen by ELISA, suggesting they had adapted to replication in these cells (Appendix A). The specimens that tested positive included two (of 15) G1P[8], four (of 11) G3P[8], one (of one) G9P[8], eight (of 14) G12P[8], and three (of 9) G2P[4] rotaviruses.

To adapt the human rotaviruses to continuous monkey kidney cell culture, we used a subset of 14 of the 18 rotavirus-positive lysates from RhMK cells representing all five genotypes as inocula for a passage series in MA104 cells in T25 culture flasks (Figure 1). We included the 29 rotavirus-positive lineages representing these 14 specimens. The initial inocula were 0.5 mL of trypsin-activated, rotavirus-positive primary RhMK lysates. In subsequent passages, we used 10 µL–1 mL of lysate as inoculum, based on ELISA score. We also passaged simian laboratory strain SA11 and medium containing trypsin as controls. For most specimens and lineages, ELISA scores tended to increase over the passages (Appendix A). Following ten passages in MA104 cells, 25 lineages representing 13 distinct specimens tested positive for rotavirus antigen by ELISA, suggesting they had adapted to replication in these cells (Appendix A). Positive specimens included one (of one) G1P[8], three (of three) G3P[8], one (of one) G9P[8], five (of six) G12P[8], and three (of three) G2P[4] rotaviruses.

We also attempted to adapt the human rotaviruses to Vero cells, which are monkey kidney cells approved for vaccine manufacture. Our reasoning was that identification of sets of mutations that allow human rotaviruses to replicate efficiently in Vero cells could be used in the future to rationally design attenuated vaccine strains that retain many antigenic epitopes of circulating pathogenic human rotaviruses, but replicate more efficiently. To adapt the human rotaviruses, we first used a subset of seven rotavirus-positive lysates from RhMK cells as inocula for a passage series in Vero cells in T25 culture flasks with a methodology identical to that used for MA104 cell passage (Figure 1). Although ELISA scores for about half of the passaged lysates were rotavirus positive after P4, by P10 all were negative (Appendix A). We hypothesized that adaptation to MA104 cells might promote adaptation to Vero cells. Therefore, we used MA104 lysates from rotavirus positive specimens and lineages as inocula for a passage series in Vero cells. The initial inocula were 1 mL of trypsin-activated, rotavirus-positive P6 or P10 MA104 lysates. In subsequent passages, we used 10 µL–1 mL of lysate as inoculum, based on the ELISA score. We also passaged simian laboratory strain SA11 and medium containing trypsin. For most specimens and lineages, ELISA scores in early passages were positive, possibly due to the presence of residual rotavirus in diluted inocula (Appendix A). ELISA scores tended to decrease over the passages, and none were positive by P10 (Appendix A). Thus, the human rotavirus clinical specimens in our collection failed to efficiently adapt to Vero cells under the given conditions after primary RhMK passage or primary RhMK and MA104 cell passage.

### 3.2. Late-Passage Human Rotaviruses Replicate More Efficiently Than Some Early-Passage Viruses

To directly assess whether serially passaged human rotaviruses had adapted to MA104 cells, we compared the replication efficiencies of early- and late-passage viruses for several specimens. We adsorbed MA104 cells with trypsin-activated rotaviruses in P3 or P10 lysates at an MOI of 0.01 FFU/cell. After adsorption, we washed to remove unbound virus, then incubated the cells for 24 or 48 h and quantified virus yield. After 10 passages, the replication efficiency of laboratory strain SA11 was not statistically or appreciably different than that of a pre-passage virus stock at 24 or 48 h (Figure 2). For G1P[8], G9P[8], and G2P[4] human rotaviruses at 48 h, titers of P10 viruses were significantly higher than those of P3 viruses. Although the numbers did not reach statistical significance for G3P[8] and G12P[8] human viruses, likely due to the spread of data points from the many lineages and specimens, at 48 h, titers of P10 viruses were appreciably higher than those of P3 viruses. Consistent with increasing ELISA titers, these observations suggest that human rotaviruses of all genotypes adapted to MA104 cells during serial passage.

### 3.3. Human Rotaviruses Acquire Polymorphisms in VP4 during Serial Passage

In published studies, adaptive mutations in human rotaviruses have been detected following 30–60 passages [47,48,49,50]. However, after three to five passages in RhMK cells and 10 passages in MA104 cells, ELISA scores were high, and replication assays suggested the capacity of the viruses to replicate in MA104 cells had substantially improved (Appendix A and Figure 2). Therefore, we decided to determine whether sequence changes had arisen in the genomes of passaged human rotaviruses. To determine the genome sequences of MA104-adapted viruses, we isolated RNA from P10 stocks of 24 lineages representing 13 distinct specimens that tested positive for rotavirus antigen by ELISA, constructed libraries, and used Illumina next-generation sequencing. We also extracted and sequenced RNA from mock-infected MA104 cell lysates and P10 SA11-infected MA104 lysates as controls. When we aligned the resulting viral sequences to the reference genomes (Appendix A), we found that for the human rotaviruses, sequence coverage was consistently high for g4, which encodes VP4, and g11, which encodes NSP5 and NSP6 (Appendix A). Coverage for the remaining segments was highly variable and often quite low. However, we obtained high sequence coverage for all segments of SA11, which was prepared using the same method and passaged at similarly high titers (Appendix A). Both g4 and g11 from adapted human rotaviruses contained synonymous and nonsynonymous polymorphisms. However, since g4 sequence coverage was high for all human rotaviruses, and VP4 is an important tropism determinant, we focused our subsequent analyses on this segment and particularly on polymorphisms encoding amino acid mutations.

To identify genetic polymorphisms associated with adaptation to cultured cells, we conducted variant-calling analysis with LoFreq [58]. This approach will identify differences from the reference sequence detected at varying frequencies, not just those that have become fixed in the population. In some cases, we detected polymorphisms at a lower frequency, but the majority were detected at frequencies >95%. While changes were detected throughout the genome, the highest concentration of variants was in g4, with 168 nucleotide polymorphisms encoding 41 amino acid changes in the VP4 protein per sequenced lineage, on average (Table 1 and Appendix A). G2P[4] virus populations contained the highest numbers of g4 variants, and most P[8] virus populations contained similar numbers of polymorphisms (Appendix A). While variants were detected in each of the P10 SA11 segments, the number in SA11 g4 (17) was very low relative to the numbers in the human P10 rotavirus g4 segments, and it was unremarkable compared to the number of variants detected in any other P10 SA11 segment. No viral sequences were detected in control P10 cell lysates. Together, these findings suggest that human rotaviruses of all genotypes serially passaged in RhMK cells then MA104 cells acquired polymorphisms in g4, some of which encoded VP4 amino acid changes. We were unable to confidently assess the changes acquired in most other segments.

### 3.4. Putative VP4 Adaptive Polymorphisms Are Conserved across Genotypes

In previous studies, amino acid mutations in the VP4 protein have been associated with tissue culture adaptation [47,48,49,50]. We identified several nonsynonymous nucleotide polymorphisms in g4 of P10 MA104-passaged human rotaviruses (Table 1 and Appendix A). Some of these were located at positions that had been identified in prior studies, although many were not (Table 2 and Appendix A). We rationalized that if human rotaviruses share a common mechanism of tissue culture adaptation, we might identify VP4 mutations that are conserved for adapted human rotaviruses across genotypes. So, we compared the identities of VP4 mutations in MA104 P10 human rotaviruses and the frequency with which they were detected among the virus lineages. Since we only had a single sequenced rotavirus-positive MA104 P10 G1P[8] rotavirus, all detected VP4 mutations were present in 100% of lineages (Table 1 and Appendix A). For the other genotypes, we calculated the frequency of mutation detection among individual sequenced lineages. Among the P[8] viruses, we identified 28 VP4 mutations that were conserved in at least half of all sequenced lineages (Table 2). Of these 28 VP8 mutations, 15 were detected in 100% of sequenced lineages. Seven of the 28 mutations in P[8] rotaviruses (S78I, G145S, V390L, V580I, A587I, V604L, and T738I) were encoded by two nucleotide polymorphisms for at least some genotypes, while the rest were encoded by a single nucleotide polymorphism (Appendix A). Among all five rotavirus genotypes, including both P[8] and P[4], we identified 12 VP4 amino acid polymorphisms that were conserved in at least half of all sequenced lineages. Four of the mutations in P[4] rotaviruses (Y295F, D385H, F467L, and V604L) were encoded by two nucleotide polymorphisms, while the rest were encoded by a single nucleotide polymorphism (Appendix A). For each given mutation, the frequency of detection of the variants encoding the mutation among sequenced reads for each lineage was >99% (Appendix A). For each genotype, we identified additional mutations in VP4 encoded by sequence polymorphisms detected in more than 50% of lineages, some of which were conserved among a subset of genotypes (e.g., Y19H) and some of which were unique (e.g., I130V) (Table 2). Polymorphisms conserved among P[8] viruses map to four main regions of VP4, (i) the VP8* head domain, (ii) within or adjacent to the VP5* hydrophobic loops, (iii) clustered around the VP5* ‘waist’, and (iv) within the VP5* foot (Figure 3A). Polymorphisms conserved among all five rotavirus genotypes map primarily to the three regions of VP5* described above (Figure 3B–D). In the reversed conformation, mutations located in the VP4 waist can be seen adjacent to the VP7 layer of the capsid, and the ring of mutations in and adjacent to the hydrophobic loops is discernable, whereas these clusters are less obvious in the upright conformation of VP4. Together, these observations reveal multiple polymorphisms encoding VP4 amino acid mutations arising during serial passage, a subset of which is conserved across genotypes. Recurrent amino acid changes appearing independently in lineages of different genotypes may contribute to tissue culture adaptation via a common mechanism.

## 4. Discussion

We serially passaged supernatants of rotavirus-positive stool samples in RhMK cells and MA104 cells. We successfully adapted rotaviruses from genotypes predominantly associated with human disease, including G1P[8], G3P[8], G9P[8], G12P[8], and G2P[4]. For P[8] rotaviruses, genome segments other than those encoding the outer-capsid proteins typically belong to the same genogroup and are more genetically similar to one another than those of G2P[4] rotaviruses, whose segments belong to a distinct genogroup [63]. Thus, it is not surprising that many more mutations were shared among the P[8] rotaviruses in our study than between a given P[8] rotavirus and the G2P[4] rotaviruses (Table 2). Nonetheless, the identification of a subset of VP4 mutations conserved across all five genotypes suggests there may be shared mechanisms of tissue culture adaptation for rotaviruses. We attempted to passage lysates from both RhMK and MA104 serial passages in Vero cells but were unable to adapt the lysates under the conditions used. Thus, in some cases, culture adaptation mechanisms may be cell line specific. Accordingly, in published studies, human rotavirus strain Wa (G1P[8]) acquired different sets of adaptive mutations following serial passage in distinct monkey kidney epithelial cell lines (Appendix A) [48,49].

Additional studies are needed to define the steps at which infectivity may be enhanced by VP4 mutations that are conserved across all genotypes. However, their predicted locations based on structures of CDC-9 in upright and reversed conformations on rotavirus particles provide insights that allow some speculation (Figure 3) [35,51]. Mutations in VP5* residue D385 have been identified in multiple culture-adapted human rotaviruses, including 89-12, CDC-9, M, and Wa, all of which are P[8] rotaviruses [47,49,50,64]. D385 is located adjacent to one of the hydrophobic loops that interacts with the lipid bilayer during rotavirus permeabilization of the endosome [36]. While mutations that reduce the hydrophobicity of these loops inhibit viral entry [65], it is unclear how a charge-altering mutation adjacent to one of the loops influences this process. Nonetheless, detection of D385H in VP4 in the majority of successfully passaged G1P[8], G3P[8], G9P[8], G12P[8], and G2P[4] rotaviruses in the current study underscores the importance of this residue for tissue culture adaptation (Table 2). In addition to D385, we identified mutations at conserved amino acid positions Y295, S383, and V390 for most passaged rotaviruses of all genotypes, which also are predicted to reside in or adjacent to the hydrophobic loops (Figure 4B). V390L within one of the loops is predicted to retain its hydrophobicity, and Y295F will retain its bulkiness. However, S389R will introduce a charged rather than polar residue near the hydrophobic loops, which could potentially influence interactions with membranes. Adaptive mutations in residues 331 and 388 in the hydrophobic loops have been identified for other human rotaviruses, further highlighting the importance of this region (Appendix A) [47,48,49,50].

Other conserved VP5* mutations are predicted to localize to the foot and waist regions (Figure 3). The VP5* foot is buried in the VP6 capsid layer in the upright conformation, and its structure in the reverse conformation is unknown, likely because it is unstructured [35,51]. Conserved mutations in the foot domain at V604 and K621 largely maintained their charged or hydrophobic character following adaptation, suggesting only fine-tuning of molecular properties is likely to result from the changes (Table 2). However, it is possible that mutation of K621 alters other functions, such as lysine-linked ubiquitination. Although the specific functions of mutations in the waist region are unknown, several have been detected in prior studies of adapted human rotaviruses, including at residues 262, 267, 268, 364, 368, 471, and 474, suggesting an important role for this region in adaptation (Appendix A) [48,49,50]. For CDC-9, an S331F mutation in VP5* has been proposed to stabilize the upright conformation of VP4, thereby enhancing infectivity [51]. Interestingly, some of the conserved mutations we detected in the VP5* waist might destabilize inter-subunit interactions in the upright conformation. For example, R268 or its equivalent residue in one monomer of VP5* interacts with VP7 in upright RRV and CDC-9 VP4 structures (Figure 4C) [35,51]. In the upright VP4 conformation of CDC-9, D252 and R268 from different VP5* monomers appear to interact ionically (Figure 4D) [51]. In either case, the conserved R268S mutation we detect in adapted human rotaviruses might reduce the stability of interactions with adjacent residues, permitting easier triggering from the upright to reverse conformation (Table 2). It is possible that in addition to having more VP4 molecules in the upright conformation, as proposed by Jenni et al. [51], the capacity to more easily transition to the reversed conformation following interaction of the hydrophobic loops with membranes also increases rotavirus replication efficiency. Adaptation might involve fine adjustments to achieve an optimal metastable state of VP4 that balances binding efficiency with fusogenic conformational transitions.

VP8* is the receptor binding domain of VP4. Although we identified few VP8* polymorphisms that were conserved for both P[4] and P[8] genotypes, several were conserved among P[8] genotypes (Table 2). Mutations we identified in the VP8* head domain of P[8] rotaviruses are not located in known glycan-binding sites [27,66,67,68,69]. Alignment of CDC-9 VP8*, onto which locations of conserved P[8] mutations we identified had been mapped, with the structure of a human P[8] VP8* domain bound to the secretory H type-1 antigen, shows that the mutations are distinct from the ligand binding site (Figure 4E) [66]. Residues 145 and 150 are located near the sialic acid binding cleft of some animal rotavirus strains but are not critical for sialic acid binding [27,67]. Thus, detected VP8* mutations are not expected to directly alter the glycan binding capacity of the virus. Conserved TNFR-associated factor (TRAF) binding motifs have been identified in VP4 and VP8* that bind TRAFs and increase in NF-κB activity [70]. Mutations we identified in the VP8* head domain of P[8] rotaviruses are not located in the two conserved TRAF binding motifs that have been identified for human rotaviruses. Previous studies of adapted human rotaviruses have also identified mutations outside of receptor binding regions, including near the base of VP8*, adjacent to VP5*. These include residues 51, 52, 77, 79, 167, 205, and 205 (Appendix A) [47,48,49,50].

There are several caveats to our findings. In our passage series, we initially inoculated cells with dilutions of stool specimens based on volume rather than virus concentration. Based on the range of ELISA A_450_ values, which are available for stools collected from 2010–2013, specimens varied widely in virus concentration (Appendix A). While MOI could influence adaptation, a low ELISA A_450_ value in the inoculum did not preclude adaptation, nor did a high value guarantee successful adaptation in our study. After the initial passages, we made some adjustments to inoculum volume based on the outcome of ELISAs. However, more frequent ELISAs and higher inoculum dilutions between passages might have reduced the effective MOI, allowing for fewer coinfections and increasing selection for individual viruses that replicate efficiently in non-human cells. Reasons for low sequence coverage across regions of the adapted human rotavirus genomes are unclear. We think that problems with our library preparation or the sequencing run are unlikely, since coverage for SA11 is excellent for all segments (Appendix A). Passaged human rotavirus libraries were prepared and sequenced alongside the SA11 library using identical protocols. Adjustments to the SA11 protocol may be needed for efficient library preparation for human rotaviruses. However, differences in %GC content do not necessarily explain virus- or segment-specific differences (Appendix A). In many cases, SA11 segments have slightly higher GC content than human rotaviruses. Segments that had good sequence coverage have similar GC content to other segments that had poor coverage. Instead, differences in the quantities of segments present, their stability, or the efficiency of random priming of specific segments or regions might have caused the observed differences in coverage. Our initial rationale for using Illumina sequencing followed by analysis with LoFreq was that we would achieve coverage of the entire genome with depth that would enable sensitive detection of polymorphisms. We reasoned that if identical polymorphisms arose in multiple lineages, even if they had not become fixed in all lineages, their presence might indicate that they were biologically meaningful. In the end, we achieved deep coverage only for g4 and g11, and most conserved polymorphisms in g4 were present in the population at an extremely high frequency. VP4 is a key rotavirus receptor-dependent tropism determinant [23,24,30], but it is possible that synonymous or nonsynonymous changes at other loci contribute to the enhanced replication we observed for human rotaviruses following serial passage (Figure 2). For example, CDC-9 exhibited significantly reduced STAT-1 activation following adaptation to Vero cells and contained amino acid mutations in proteins other than VP4, including a single mutation in innate immune antagonist protein NSP1 [50]. Culture-adapted CDC-9 replicated to higher titers than the parent virus in Caco-2 and Vero cells. While we failed to detect an identical mutation in NSP1 of any of our passaged viruses adapted to MA104 cells, sequence read counts were low in NSP1-encoding g5 for most adapted viruses (Appendix A). Other potential mechanisms of adaptation involve enhanced viral replication or spread. Indeed, the greater difference in replication at 48 h than 24 h for several P10 human rotaviruses compared with P3 (Figure 2) suggests that these viruses may have adapted more efficiently to some aspect of replication or spread in culture following infection, rather than simply enhancing attachment or entry. In the current study, we passaged human rotaviruses three to five times in primary RhMK cells, then 10 times in MA104 cells. Adaptation has historically been initiated in primary AGMK cells, which may yield different results or promote adaptation to Vero cells [48,62]. Nonetheless, our results suggest primary RhMK cells are suitable for human rotavirus adaptation, at least to MA104 cells. For some human rotaviruses, greater passage numbers have been used to identify adaptive polymorphisms or generate vaccine candidates [47,48,50]. With additional passages, a subset of critical adaptive polymorphisms might have become fixed in our virus populations.

Poor replication of contemporary human rotaviruses in cultured cells is an important deterrent to bespoke rotavirus vaccine engineering, and VP4 is a primary rotavirus tropism determinant. In the current study, we were unable to adapt human rotaviruses to Vero cells, which are used for rotavirus vaccine manufacturing. Nonetheless, the detection of conserved polymorphisms upon human rotavirus adaptation to MA104 cells suggests there may be conserved, genotype-independent mechanisms of tissue culture adaptation that can be identified in future studies and used towards this end. The recovery of human rotaviruses and animal rotaviruses containing human outer-capsid antigens by reverse genetics systems underscores their potential as future vaccine platforms [38,39,71,72,73]. A combination of structural and functional analyses that involve the use of rotavirus reverse genetics systems may allow us to elucidate fundamental mechanisms by which polymorphisms acquired during serial passage enhance rotavirus replication.

## Figures and Tables

**Figure 1 viruses-16-00978-f001:**
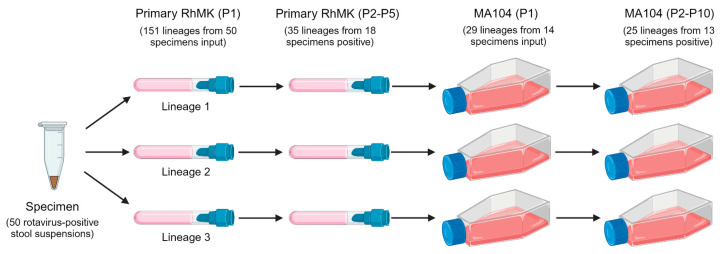
Workflow for rotavirus serial passaging. Each human rotavirus clinical specimen was serially passaged in triplicate lineages in primary RhMK cells in roller tubes three to five times. Most rotavirus-positive lineage lysates from RhMK passage were serially passaged ten times in monkey kidney epithelial MA104 cells in tissue culture flasks. Numbers of input lineages and specimens in each series and of rotavirus positive lineages and specimens at the end of each series are indicated.

**Figure 2 viruses-16-00978-f002:**
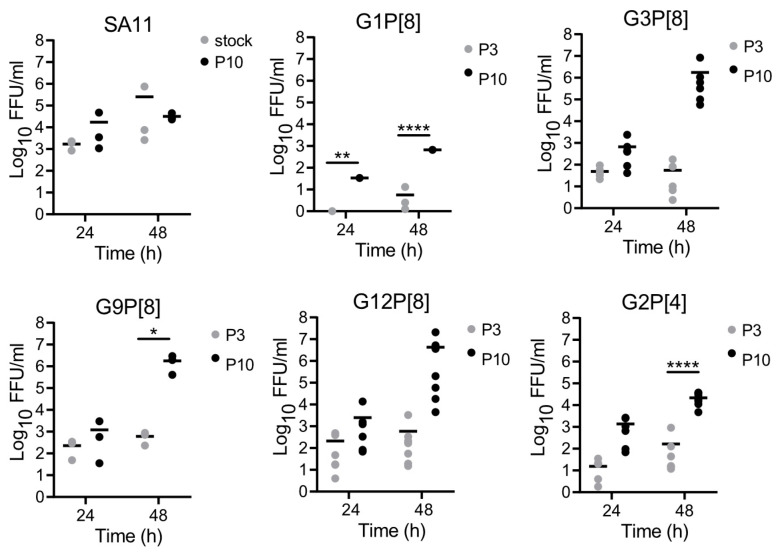
Replication of early- and late-passage rotaviruses in MA104 cells. MA104 cells were adsorbed for 1 h at 37 °C with trypsin-activated, rotavirus-positive P3 or P10 lysates at an MOI of 0.01 FFU/cell or with undiluted lysate if an MOI of 0.01 FFU/cell could not be reached. Cells were washed to remove unbound virus and incubated with serum-free medium plus 0.5 µg/mL trypsin at 37 °C for 0, 24, or 48 h. Plates were frozen at −80 °C and thawed three times prior to determining virus titer by FFA on MA104 cells. The mean and individual data points are shown. Virus yield was determined by dividing titer at 24 h or 48 h by titer at 0 h. *, *p* < 0.05; **, *p* < 0.01; ****, *p* < 0.0001 by two-way ANOVA with Šídák’s multiple comparisons.

**Figure 3 viruses-16-00978-f003:**
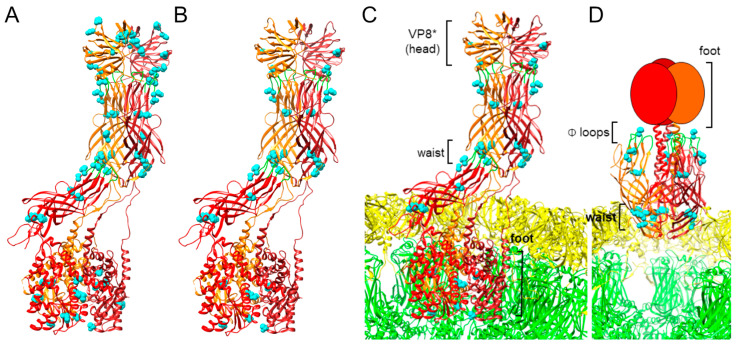
Locations of conserved VP4 amino acid mutations. VP4 of human G1P[8] vaccine candidate CDC-9 is shown in an upright ((**A**–**C**); PDB ID 7UMS) or reversed ((**D**); PDB ID 7UMT) conformation [51]. In (**C**,**D**), VP4 is shown relative to VP7 (yellow) and VP6 (green) rotavirus capsid layers. Monomers of trimeric VP4 are shown as red and orange ribbons, with hydrophobic loops colored green, or as red and orange ovals when the structure is unresolved. Locations of VP4 polymorphisms detected following serial passage in MA104 cells that are conserved for P[8] rotaviruses (**A**) or across all tested genotypes (**B**–**D**) are shown as spheres and colored cyan.

**Figure 4 viruses-16-00978-f004:**
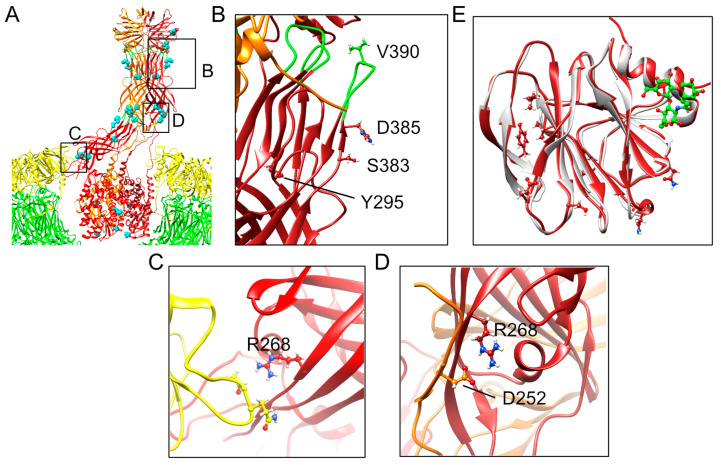
Locations and potential interactions of conserved VP4 amino acid mutations. VP4 of human G1P[8] vaccine candidate CDC-9 is shown in an upright conformation relative to VP7 (yellow) and VP6 (green) rotavirus capsid layers ((**A**); PDB ID 7UMS) [51]. Areas that are magnified in panels (**B**–**D**) are indicated. Monomers of trimeric VP4 are shown as red and orange ribbons, with hydrophobic loops colored green. Locations of VP4 polymorphisms detected following serial passage in MA104 cells that are conserved across all tested genotypes are shown as cyan spheres in (**A**) or as ball-and-stick representations in (**B**–**E**). In (**E**), the VP4 head domain of CDC-9 (dark red) onto which locations of conserved VP4 mutations from this study have been mapped, has been aligned with a human P[8] VP8* domain (light gray) bound to the secretory H type-1 antigen (green) (PDB ID 6HA0) [66].

**Table 1 viruses-16-00978-t001:** Variant summary.

Genotype	G1P[8]	G3P[8]	G9P[8]	G12P[8]	G2P[4]
Specimens	1	3	1	5	3
Total lineages	1	6	3	7	7
Average total variants	315	224	319	291	837
Average mutation frequency	0.019	0.020	0.027	0.042	0.023
Average g4 variants	152	149	178	176	183
VP4 amino acid changes	35	42	37	39	54

**Table 2 viruses-16-00978-t002:** VP4 amino acid changes acquired during serial passage by genotype.

**G1P[8]**	**G3P[8]**	**G9P[8]**	**G12P[8]**	**G2P[4]**		**G1P[8]**	**G3P[8]**	**G9P[8]**	**G12P[8]**	**G2P[4]**
-	Y19H ^1^	Y19H	Y19H	-		-	-	A430T	-	-
H52Y *	H52Y *	H52Y *	H52Y *	-		-	-	-	-	I439L
T78I	S78I	S78I	T78I	-		-	-	-	-	M444V
-	-	-	G/N99S	-		-	-	-	-	V463I
I106V	I106V	I106V	I106V	-		** F467L **	** F467L **	** F467L **	** F467L **	** F467L **
V108I	V108I	V108I	V108I	-		-	-	-	-	N498T *
-	D113N	D113N	-	-		S546N	-	S546N	-	-
N120T	N120T	N120T	N120T	-		V560I	V560I	V560I	V560I	-
-	I130V	-	-	-		-	-	-	-	A578V
G145S	G145S	G145S	G145S	-		V580I	V580I	V580I	V580I	-
-	-	-	T149N	-		-	K581R	-	-	-
D150E	D150E	D150E	D150E	-		-	-	-	-	L584I
R162K	R162K	R162K	-	-		-	A586T	A586T	-	-
-	-	-	-	M166I		V587I	V587I	V587I	A587I	-
V173I	-	-	-	V173I		** W590L **	** W590L **	** W590L **	** W590L **	** W590L **
-	-	-	-	S189N		-	-	-	-	S591T
-	-	-	-	D192N		-	-	-	-	D592N
-	S194N	-	-	-		I593V *	-	-	-	A593V *
** G195D ^2^ **	** G195D **	** G195D **	** G195D **	** N195D **		-	-	-	-	K595N
T199I	T199I	T199I	T199I	-		-	-	-	-	S596D
-	-	-	-	R245K		-	-	-	-	L598S
** D252E **	** D252E **	** D252E **	** D252E **	** D252E **		-	S599N	-	-	-
** R268S * **	** R268S * **	** R268S * **	** R268S * **	** R268S * **		-	-	-	L600V	-
-	-	-	-	V280I		-	-	-	-	D602N
-	I281V	I281V	I281V	I281V		** V604L **	** V604L **	** V604L **	** V604L **	** V604L **
** Y295F **	** Y295F **	** Y295F **	** Y295F **	** Y295F **		-	A608S	A608S	-	-
-	-	-	-	S303N		-	-	-	R616K	-
-	-	-	-	S305L		N617K	N617K	N617K	S617R	-
V338I	-	-	-	-		** K621R **	** K621R **	** K621R **	** K621R **	** K621R **
-	-	-	-	I352V		-	-	-	-	I629M
** A360V **	** A360V **	** A360V **	** A360V **	** A360V **		-	A642T	-	-	-
T380A	T380A	T380A	T380A	-		-	-	-	-	V674I
** S383R **	** S383R **	** S383R **	** S383R **	** S383R **		-	-	-	-	V683I
** D385H * **	** D385H * **	** D385H * **	** D385H * **	** D385H * **		F689V	F689V	F689V	F689V	-
-	-	-	-	R387S		-	-	-	-	I704V
I388L *	I388L *	I388L *	I388L *	-		T708A	-	T708A	-	-
** V390L **	** V390L **	** V390L **	** V390L **	** V390A/L **		I711V	-	-	-	-
-	-	-	-	E392A		-	-	-	-	D713N
-	-	-	-	I395V		T738I	T738I	T738I	T738I	-

^1^ Amino acid mutations were included in the table if they were detected in a minimum of 50% of sequenced lineages for a given genotype. ^2^ Bold, underlined text indicates amino acid mutations that are conserved for at least 50% of sequenced lineages per genotype for all five adapted rotavirus genotypes. * Position identified as an adaptive mutation in a prior study [47,48,49,50].

## Data Availability

Data generated from Illumina RNA-seq can be accessed at the NCBI Sequence Read Archive (SRA) under BioProject accession number PRJNA1100611. Code utilized in this report can be accessed at https://github.com/ogdenlab1/OrthoreoVariant accessed on 6 June 2024.

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
