# Peer review of "Human Rotaviruses of Multiple Genotypes Acquire Conserved VP4 Mutations during Serial Passage"

_viruses, 2024, doi:10.3390/v16060978_

Round 1

Reviewer 1 Report

Comments and Suggestions for Authors

The submitted manuscript by Carter and colleagues describes studies to identify rotavirus cell culture entry-licensing mutations that result in successful replication. These evolutionary changes define a trajectory of adapting epidemiologically relevant clinical RV strains for vaccine development. The results shown identify several conserved HuRV determinants of culture adaptation. A key innovation here is the use of primary RhMK (vs. AGMK) cells for adaptation, and this seems to result in adaptation that is cell line specific, potentially explaining why similar efforts in AGMK cells did not prove to be as successful. The authors also characterized multiple human genotypes associated with severe disease in the clinic using Illumina, which is an improvement over prior approaches. The integration of genomic and structural insights to derive conclusions on VP4 functions related to adaptation in culture is excellent, and helps assimilate the results in a broader context.   

There are some drawbacks in the study - I am not listing these in any particular order: initiating adaptation to culture under different MOIs could mean that tropism is (also) determined by factors that restrict RV spread and not entry, the github resource provided to reproduce the findings here is a key resource and needs to be carefully curated, sequence coverage was inadequate for all RV genes except VP4./VP7 and g11 and it is not totally clear why this happened (as explained), non-cell entry/VP4/VP7 tropism mechanisms could partly drive adaptive mutations identified and needs to be discussed adequately. 

Most concerns can be addressed by the authors without additional experiments. Overall, I find this study to be interesting and relevant. The VP4 amino acids conserved across multiple human RV genotypes involved in adaptation of clinical human RV isolates will be useful to better understand RV entry determinants (and rationally attenuate vaccine strains). These findings should be followed up in future using reverse genetics and structure-function validation in pre-clinical models. 

Specific comments:

Major comments:

  1. MOI and non-RV cell attachment/entry drivers of culture adaptation: The initial rounds of culture adaptation are done using a 1:10 diluted stool specimen across all 46 clinical isolates. This presumably initiates adaptation of the 46 clinical specimens at different MOIs, which could be important for RV replication and spread in a manner independent of VP4/VP7 attachment mechanisms. Is there any data available on RV antigen levels in these stool specimens (P0) that could help in assessing their relative viral loads? Similarly, when the n=14 RhMK-adapted specimens are initiated in MA104 cells a uniform 0.5-mL inoculum is used (rather than an ELISA-normalized inoculum volume) and similar differences in MOI could be expected, determining virus spread (and eventual ‘adaptation’ in an attachment-independent manner). This is also the case for MA104 lysates initiated in Vero cells (lines 278:280). Although there is no easy way around this bottleneck without full-genome mapping, some discussion is warranted in the paper of MOI and virus spread as additional potential factors determining successful culture adaptation of the clinical isolates.  

  2. L559-561: data availability: the github code resource link provided seems to be totally unrelated to this manuscript (instead related to coronaviruses?) and needs to be overhauled - as examples, these scripts in the github rep from the link provided seem to be totally unrelated to the manuscript: 

Chimeric Recombination/MERS-SARS2_Junction_summary.R. 

Chimeric Recombination/sgmRNA_filtering_MERS-SARS2_chimeras.R

The authors should carefully set up a github rep for the manuscript and modify their manuscript link. Ideally this will include the scripts needed to ingest and analyze the RNA-seq data as done in this paper, as well as allow a reader to generate the summary tables shown in supplementary information. I could not access the bioinformatics methods for the figures and tables presented and thus cannot verify these until this issue is addressed. Also see my comment below on providing parsable data for supplementary tables as tsv/csv files instead of PDF documents. Minor comments (listed by line numbers or sections):

Section 3.2: replication efficiency of a simian SA11 strain in the simian MA104 cell line did not increase after 10 passages compared to the human RVs tested. In this case it would have been interesting to know whether a laboratory adapted human RV strain showed a similar increase in replication efficiency - are there any prior studies from the authors’ lab that may have already addressed this possibility (or citations). Figure 1 should also be cited in this section. 

L48-49: “only a subset of G/P type combinations infect and cause disease in humans [16].” suggest changing this to “subset of G/P type combinations can infect and/or cause disease in humans with varying degrees of severity”. Not all G/P type combinations capable of infecting humans cause disease severity that is comparable - likely due to the effects of non-cell attachment virulence factors. 

L56: it is important to define tropism as used in the paper i.e., receptor-dependent restriction of viral replication. There are extensive studies identifying receptor-independent mechanisms of tropism for rotaviruses both in vitro and in vitro.

L107: ‘insensitive Sanger sequencing’ - some clarification/citations needed here on the specific merits of Illumina/deep sequencing in detection of viral mutants that could go undetected using Sanger approaches. Some of the supplementary data showing the ability of Illumina to uncover rare frequency mutations can be discussed in support of this argument.

L258 and L270: the frequencies of positive detection for genotypes should be included (e.g. 2/16 G1P[8], 1/1 G9P[8], etc). 

L299-303: It is interesting that for some of the genotypes tested, the difference between early and late passaged virus is greater at 48 hpi than at 24 hpi (i.e., P3/P10 delta at 24hpi > 48hpi). This suggests that late passage strains are better adapted to spread in culture following initial infection, rather than attach. The authors should comment on this finding. 

L321-324: it is not totally clear to me that recombination events/abnormal gene segments could influence sequencing coverage since g11 is particularly prone to this kind of rearrangement but showed good coverage. 

L329-345: Considering polymorphism frequencies (for g4) for a particular genotype across multiple lineages adapted to culture - this is described in the next section but it would be useful to include information on lineage polymorphism frequency by genotype in the summary table 1. 

L412-413: suggest changing “culture adaptation mechanisms may be cell type specific” to “cell-line specific” as the cells referred to are the same cell type (kidney epithelial cells). 

L420-421: as written this statement needs some additional wording. A possible mechanism for the effect of the D385H charge disrupting substitution in a hydrophobic loop could be an effect on membrane permeabilization, post-attachment (as published in several earlier papers from Harrison and colleagues) - this can be discussed in the context of D385 and other frequent adaptive mutations in this region of the protein. 

L444-446: the K621 mutation is conserved and could also reflect other functions for this residue (such as lysine linked ubiquitination). It seems that given the conserved status of this amino acid change during adaptation, it is a convergent evolutionary change whose role extends beyond fine-tuning; this can be explored in future studies. 

L428: a typo in this line should be corrected

Figure 3: this is a nice addition to the discussion and the relevance of the findings.

L455: table 5 is cited but not shown - should this be table 2 instead?

L462-471: are any of the conserved VP8* adaptive substitutions likely to disrupt VP4 signaling via TRAF-binding motifs reported earlier? The authors should check this and comment.

L473-476: as written is unclear to me - the authors need to rephrase this to improve readability. 

L476-481: as mentioned above, this should also have affected g11, a frequently rearranged gene segment (a lot of historical data - from Dusselberger, McCrae, etc that showed this). More likely that the lack of sequence coverage is due to a technical issue.  

L486-488: do the authors mean they failed to detect an identical NSP1 mutation in their Vero cell adaptation experiments? Otherwise it is unclear why this (presumably) loss-of-function in NSP1 would occur in primary cells with an intact interferon response. The authors should review this statement and modify it as appropriate.

L488-490: this sentence is unclear as written and can be removed without affecting the quality of the discussion. 

L497-513: this section seems too speculative - consider deleting altogether or shortening to improve the impact of the results. 

L559-561: data availability: the github code resource link provided needs to be overhauled. 

Supplementary tables of data should be provided as tabbed excel sheets as readers interested in parsing the datasets further will use this format rather than a pdf table. 

Table S1 lists the >30 replicate lineages of the RV strains adapted to cell culture. For readers to understand these findings better, a more formal definition of ‘lineage’ initially would be helpful (maybe in the methods). 

Table S8: adding the protein coding assignments for g1-g11 to this table would help readers identify the frequencies of mutations for a specific gene of interest. Interesting increase in variants for g1-g3, g6, g8, g10 in G2P[4] and accounting for the smaller size of g11 it appears that the frequency of adaptive mutations in this non-structural protein per site is rather high. 

Methods:

Were the cell lines used verified (STR, etc) and mycoplasma tested - if so this information should be added. The media conditions used for initial adaptation in RhMK cells (including FBS levels) are mentioned as proprietary to the manufacturer - is there additional information on the medium purchased (Cat#, etc) that can be included in the methods?

L152: diluent used for initial virus inoculum preparation should be detailed.

Reviewer 2 Report

Comments and Suggestions for Authors

Overall, this is a well written paper in general with new information on VP4 adaption from clinical isolates to cell culture. I just have a few suggestions to improve the manuscript.

Major:

1)        Are any of the Vero passage ELISA negative samples positive by QPCR? ELISA is not known for its sensitivity and Taqman probes for conserved regions of the group A rotaviruses may be able to pick up low level replicating viruses.

2)        Abstract, line 21, “12 mutations..for all five genotypes”. Only 10 mutations were in bold in Table 2.

3)        Table 2, for comparison purposes, it would be useful to include the known mutations in VP4 from the Rotarix passage experiment (from 89-12) and CDC-9 passage experiment and see if the newly identified mutations are conserved.

Minor:

1)        Why is the SA11 titer so low in Fig. 1?

2)        Line 320, it may be informative to comment on whether these mutations in gene 11 alters NSP5 and/or NSP6 protein sequence.

Reviewer 3 Report

Comments and Suggestions for Authors

The authors serially passaged 46 human rotavirus A (RVA) isolates (previously derived from clinical specimens and sequenced) in RhMK and MA104 cells. These isolates represented the five genotypes commonly associated with severe diarrheal disease in children: G1P[8], G2P[4], G3P[8], G9P[8], and G12P[8] [17]. Sequence comparison before and after serial passaging revealed the mutations associated with cell culture adaptation (with the focus on VP4 gene), of which 12 were conserved among all five genotypes. Similar studies were conducted previously by several other groups, and suggested a similar hypothesis that there may be a conservative mechanism associated with cell culture adaptation. Also, the previous findings by others suggested that some of these mutations can be associated with specific conformational changes. The authors need to acknowledge these past observations throughout when they discuss their similar conclusions based on the current study findings. They have also attempted to adapt the RVAs isolates to Vero cells, which, however, was unsuccessful. Nevertheless, this study innovation and important contribution to the research field is that it is focused on five contemporary dominant genotypes of RVA, which has not been accomplished previously. Additionally, the authors attempted to interpret the associations between the identified mutations and the predicted conformation changes in the context of the RVA VP4 transition from the ‘reversed’ to the ‘upright’ conformation following cell culture adaptation. Overall, these data are of importance; however, the manuscript writing, and the data presentation require some improvement.

One concern is that Illumina sequencing has yielded unsatisfactory coverage results for most segments. Illumina sequencing is being used routinely for RVA/RVB/RVC sequencing allowing determining complete sequences for all genomic segments. It has been successfully used to sequence wild-type and cell culture adapted RVs successfully. It is true that various genomic sequences may be present in drastically different quantities. However, I’m concerned that there was some technical error, or simply poor RNA/library preps. This should be addressed, if not Sanger or Nanopore sequencing can still provide feasible alternatives.

Please clarify why you attempted to adapt RVs from clinical samples to Vero cells, not commonly used for RV propagation/studies. What was the rationale behind this? As written, not clear.

Since much emphasis has been placed on the advantages of Illumina vs Sanger sequencing (such as detection of rare/not dominant variants and not only those that become fixed in viral population), please clarify why it is important. They may be low frequency because they reduce viral fitness/replication, and, actually, it may be counterproductive/confusing to consider those.

Please mention why P10 was selected and not a higher number passage.

Also, when you consider the past and the current research on RVA-VP4 mutations associated with cell culture adaptation, it seems there are no conserved mutations consistently identified by all studies in different RVA genotypes. So, it is likely, that the mechanisms may be redundant, and the importance of specific mutations is being overemphasized instead focusing on the region being altered and the associated conformational changes.

Some statements are not entirely accurate.

P3, L108: “…and nearly all the human RVs investigated represent a single genotype, G1P[8]”. And PP10-11, LL417-419: “Mutations 417 in VP5* residue D385 have been identified in multiple culture-adapted human rotaviruses, including 89-12, CDC-9, and Wa, all of which are G1P[8] rotaviruses [47, 49, 419 50, 70].” These statements are not accurate, ref #49 identified the D385N mutation following cell culture adaptation of two human RVAs: G1P[8] (Wa strain) and G3P[8] (M strain), and one porcine RVA G4P[6]. Please revise these sentences for accuracy as follows: P3, L108: “…and nearly all the human RVs investigated represent a single genotype, P[8]”. And PP10-11, LL417-419: “Mutations 417 in VP5* residue D385 have been identified in multiple culture-adapted human rotaviruses, including 89-12, CDC-9, M and Wa, all of which are P[8] rotaviruses [47, 49, 419 50, 70].” Apply similar changes throughout as appropriate.

P1, LL13-14: “However, most such studies 13 were conducted using Sanger sequencing and only a single rotavirus genotype”. Similar to above, the D385 mutations were found in different P genotypes of human and animal RVs

Please use RV or rotavirus/rotaviruses consistently throughout. See some examples below when you use the latter. In other instances you use the abbreviated version.

P2, LL79-80: “..the efficiency of infection, but factors that limit or enhance human rotavirus 79 infection in specific cell types are incompletely understood.”

P2, L86: “mutations have somewhat rarely been reported for human rotaviruses, 33 passages in”

P2, L89: “For human rotavirus strains Wa, DC3695, and..”

P2, L91: “Passage of human rotavirus 91 strains Wa and M in MA104 cells yielded attenuation”

Etc.

Reviewer 4 Report

Comments and Suggestions for Authors

In this manuscript, Carter et al. passaged human rotavirus clinical specimens in cultured cell lines and investigated mutations in the cell-culture adapted strains. The authors used 46 human rotavirus clinical specimens that belong to five major circulating genotypes and obtained 25 lineages of cell-culture adapted strains. Using next-generation sequencing, they identified 28 mutations conserved in P[8] genotypes, and 12 of them were also conserved in P[4] genotype. Most mutations are mapped in the VP5* domain in VP4, suggesting that the mutations are associated with conformational change of VP4 during infection rather than affecting interaction with cell surface molecules. It provides useful information for understanding rotavirus infection in cultured cell lines. The manuscript is well-written, and the data are depicted clearly. Here are some suggestions to consider.

Major comment:

1.       It would be helpful for readers to provide a schematic presentation of the serial passaging of clinical specimens with the number of positive lineages.

Minor comment:

1.       Lines 47: According to the latest edition of Rotavirus Classification Working Group, 42 G types and 58 P types are accepted as of April 3, 2023. Please see the following URL for more information.

https://rega.kuleuven.be/cev/viralmetagenomics/virus-classification/rcwg

2.       Line 289: Figure 1 is not cited in the result section.

3.       Lines 360-361 reads four of the 28 mutations but shows 7 mutations (S78I, G145S, V390L, V580I, A587I, V604L, and T738I). Please clarify.

4.       Lines 408, 446, 455, and 464: Table 5 should be Table 2.
